

# Alquimia v1.0: A generic interface to biogeochemical codes– A tool for interoperable development, prototyping and benchmarking for multiphysics simulators

Sergi Molins[1], Benjamin J. Andre[1], Jeffrey N. Johnson[1,8], Glenn E. Hammond[2], Benjamin N. Sulman[3], Konstantin Lipnikov[4], Marcus S. Day[5], James J. Beisman[3], Daniil Svyatsky[4], Hang Deng[6], Peter C. Lichtner[7], Carl I. Steefel[1], and J. David Moulton[4]

[1]Lawrence Berkeley National Laboratory, 1 Cyclotron Rd, Berkeley CA 94720
[2]Pacific Northwest National Laboratory
[3]Oak Ridge National Laboratory
[4]Los Alamos National Laboratory
[5]National Renewable Energy Laboratory
[6]Peking University
[7]University of New Mexico
[8]Cohere Consulting, LLC, Seattle, WA

**Correspondence:** Sergi Molins (smolins@lbl.gov)

**Abstract.** We present Alquimia v1.0, a generic interface to geochemical solvers that facilitates development of multiphysics simulators by enabling code coupling, prototyping and benchmarking. The interface enforces a call signature for setting up, solving, serving up output data, and other common auxiliary tasks, while providing a set of structures for data transfer between the multiphysics code driving the simulation and the geochemical solver. Alquimia relies on a single-cell approach that per-

mits operator splitting coupling and parallel computation. We describe the implementation in Alquimia of two widely-used open-source codes that perform geochemical calculations, PFLOTRAN and CrunchFlow. We then exemplify its use for the implementation and simulation of reactive transport in porous media by two open-source flow and transport simulators, Amanzi and Parflow. We also demonstrate its use for the simulation of coupled processes in novel multiphysics applications including the effect of multiphase flow on reaction rates at the pore scale with openFOAM, the role of complex biogeochemical processes

in land surface models such as the E3SM Land Model (ELM), and the impact of surface-subsurface hydrological interactions on hydrogeochemical export from watersheds with the Advanced Terrestrial Simulator (ATS). These applications make it apparent that the availability of a well-defined yet flexible interface has the potential to improve the software development workflow, freeing up resources to focus on advances in process models and mechanistic understanding of coupled problems.

## 1   Introduction

Numerical modeling has become an integral part of the investigation of some of the world's most pressing environmental challenges, such as climate change, pollution prevention, contaminant remediation or nuclear waste management (Steefel et al., 2015; Li et al., 2017). For an accurate system representation, models must consider the gamut of processes affecting mass





balances and underlying biogeochemical transformations. Different models exist for representing biogeochemical processes including aqueous complexation, mineral dissolution-precipitation, surface sorption, and microbiologically-mediated reactions.

The mathematical expressions for biogeochemical reaction models are diverse and generally express nonlinear relationships with complex parameterizations between primary variables such as concentrations (Steefel et al., 2015).

Multicomponent reactive transport codes couple biogeochemical models with solvers for flow and transport and other relevant processes such as heat transfer or geomechanics (Steefel and MacQuarrie, 1996; Steefel et al., 2005, 2015). Many of these codes are the legacy of years of development and research, a period over which model complexity has increased incrementally.

Further development is driven by continued advances in process model descriptions across spatial scales and hydrological domains, from single pores to the subsurface reservoirs or watersheds. There is also a growing need to expand the role of reactive transport to explore nonlinear interactions among the atmosphere, hydrosphere, biosphere, and geosphere in combination with other community models such as land surface models (Li et al., 2017; Sulman et al., 2024). In parallel, hardware architectures evolve continuously and new numerical approaches become available for application. Jointly, these developments enable the

simulation of tighter process coupling with increasing resolution and mechanistic detail, but they also demand continuous code development and, at times, refactoring. This increasing degree of sophistication also presents some challenges. How does one develop, test and ultimately incorporate biogeochemical capabilities in codes for new applications? How does one ensure the validity of a coupled simulator formulation prior to application? How does one continuously develop code to incorporate new numerical approaches or to take advantage of new computational resources?

Often the complexity of implementing a comprehensive and flexible treatment of biogeochemistry is a significant obstacle to the development of new biogeochemical capabilities. As a result, this step is often circumvented by coupling flow and transport codes to existing biogeochemical codes, an approach widely used, e.g. HYTEC+CHESS (van der Lee et al., 2003), Chombo-Crunch (Molins et al., 2012), Comsol+PHREEQC (Nardi et al., 2014; Jara et al., 2017), or ParCrunchFlow (Beisman et al., 2015). The use of PHREEQC as geochemical solver in this role for multiple codes, e.g. PHT3D (Prommer and Post, 2010),

Hydrus/HPx (Simunek et al., 2013) and PHAST (Parkhurst et al., 2010), has led to the development of dedicated coupling tools such as IPhreeqc (Charlton and Parkhurst, 2011) or PhreeqcRM (Parkhurst and Wissmeier, 2015). However, these tools are specific to PHREEQC and thus tied to its capabilities.

Model development entails (among other tasks) prototyping, implementing and verifying the new coupled capabilities. Prototyping makes it possible to evaluate approach feasibility and often benefits from using high-level weakly-typed languages

such as Python. Implementation requires a clear, well-documented description of the data structures and function calls that must allow enough flexibility such that a broad range of applications is possible. Ensuring the validity of complex coupled models involves intercomparison studies where multiple codes solve the same problem (Steefel et al., 2015; Maxwell et al., 2014; Molins et al., 2020).

In this manuscript we introduce Alquimia, a new open-source software library that provides a generic interface to existing

biogeochemical capabilities. This software is intended to facilitate interoperable code development by exposing tried-and-true biogeochemical capabilities in existing software. We present a general formulation of the problems the interface is designed for (Sec. 2.1), and then describe the functions and data structures of the interface along with software design approaches (Sec.





3). The implementation and use of Alquimia are given by the way of examples (Sec. 4). First, the geochemical capabilities in the open-source reactive transport codes PFLOTRAN and CrunchFlow are implemented in Alquimia. Then, these geochemical capabilities are made available for the simulation of reactive transport in porous media in two codes, Amanzi and ParFlow. We show that because Alquimia allows for different geochemical codes to share a common flow and transport solver, and therefore the same spatial discretization, time stepping control, and coupling schemes, it may be a useful tool for multi-way model intercomparison. We present examples of how Alquimia has enabled incorporation of geochemical capabilities to codes for a range of applications, including prototyping land-surface processes in Python as well as in high-performance computing simulations using an OpenFOAM-based code and the Advanced Terrestrial Simulator (ATS) (Sec. 5).

## 2 Model description

### 2.1 Mass balance equations

The mass balance of each aqueous species may be written as

$$\frac{\partial \theta c_i}{\partial t} = L(c_i) + r_i \tag{1}$$

where $c_i$ is the concentration of species $i$, $\theta$ is the volumetric water content, $L(c_i)$ represents the transport operator and $r_i$ is the contribution of reactions to the mass balance of species $i$. While $L(c_i)$ represents the solute transport operator here, it may be also read as a more general operator that includes other processes.

Equilibrium aqueous complexation reactions make it possible to rewrite Eq. 1 as

$$\frac{\partial \theta \Psi_i}{\partial t} = L(\Psi_i) + R_i \tag{2}$$

where the aqueous concentration of each component ($\Psi_i$) is defined as the sum of the concentration of a primary species and a set of $N_x$ secondary species (Steefel and MacQuarrie, 1996)

$$\Psi_i = c_i + \sum_{j=1}^{N_x} \nu_{ij} c_j \tag{3}$$

where $\nu_{ij}$ is the stochoimetric coefficient of component $i$ in reaction $j$ and $R_i$ is the contribution of kinetic reactions to the mass balance of component $i$.

This approach reduces substantially the number of governing equations and unknowns from $N_s$ species in Eq. 1 to $N_c$ components in Eq. 2 by using the mass of law action equations of the $N_x$ equilibrium reactions

$$c_j = \frac{\prod_{i=1}^{N_c} (\xi_i c_i)^{\nu_{ij}}}{\xi_j K_j} \tag{4}$$

where $\xi_i$ and $\xi_j$ are the activity coefficient of primary and secondary species, and $K_j$ is the equilibrium constant of reaction $j$.

Mass action law equations for heterogeneous equilibrium reactions follow a similar form (Steefel et al., 2015, e.g.). When these are considered, the mass of each component ($\Psi_{t,i}$) also includes the mass present as a mineral ($\Psi_{m,i}$) or sorbed/exchanged





on a surface $(\Psi_s, i)$

$$\Psi_{t,i} = \Psi_i + \Psi_{m,i} + \Psi_{s,i} \tag{5}$$

The kinetic reactions rates are calculated as functions of primary variables such as concentrations $(c_i)$ and sets of intrinsic parameters $(p_k)$ that are specific to each reaction $k$

$$R_i = \sum f_k(c_i, p_k) \tag{6}$$

The $f_k$ functions take different mathematical forms for different reaction types. Different codes may implement somewhat different formulations or allow for generalized formulations (Mayer et al., 2002, e.g.) and custom rate expressions (Hammond, 2022). Examples of established formulations include the transition-state-theory-type (TST) rate law for mineral-dissolution-precipitation or the Monod-type rate expression for microbially-mediated reduction-oxidation reactions. The interface presented in Sec. 3 does not stipulate that any specific mathematical form that is used. Hence, we do not provide a specific form for $f_k$ here. In Sec. 4.3, we compare codes when well-established reaction models are used in different geochemical engines. In Sec. 5, we give some specific reaction rate expressions connected to example applications.

## 2.2  Coupling approaches

The set of equations for all components along with the geochemical equilibrium and kinetic equations (Eqs. 2-6) (e.g., Steefel et al., 2015), is in general a coupled system of nonlinear equations. Two broad approaches are used to solve this system. The global implicit approach entails the simultaneous solution of the coupled system including reactions and transport, often directly substituting Eqs. 4, 6 in Eq. 2. In contrast, operator splitting consists of, first, simulating transport for each component (and phase) separately, and second, updating the concentrations by solving the geochemical equations. Mathematically, the method can be represented as a two-step sequential process consisting of a transport step $(tr)$

$$\frac{(\theta^{tr}\Psi_i^{tr} - \theta^n\Psi_i^n)}{\Delta t} = L(\Psi_i^n) \tag{7}$$

followed by a reaction step

$$\frac{\theta^{tr}(\Psi_i^{n+1} - \Psi_i^{tr})}{\Delta t} = R_i^{n+1} \tag{8}$$

where the reaction step includes the time-discretized form of the component balance over the different phases (Eq. 5), equilibrium aqueous speciation (Eq. 3), and kinetic reactions (Eq. 6)

$$\Psi_{t,i}^{n+1} = \Psi_i^{n+1} + \Psi_{m,i}^{n+1} + \Psi_{s,i}^{n+1}$$

$$\Psi_i^{n+1} = c_i^{n+1} + \sum_{j=1}^{N_x} \nu_{ij}^{n+1} c_j^{n+1}$$

$$R_i^{n+1} = f_k(c_i^{n+1}, p_k) \tag{9}$$





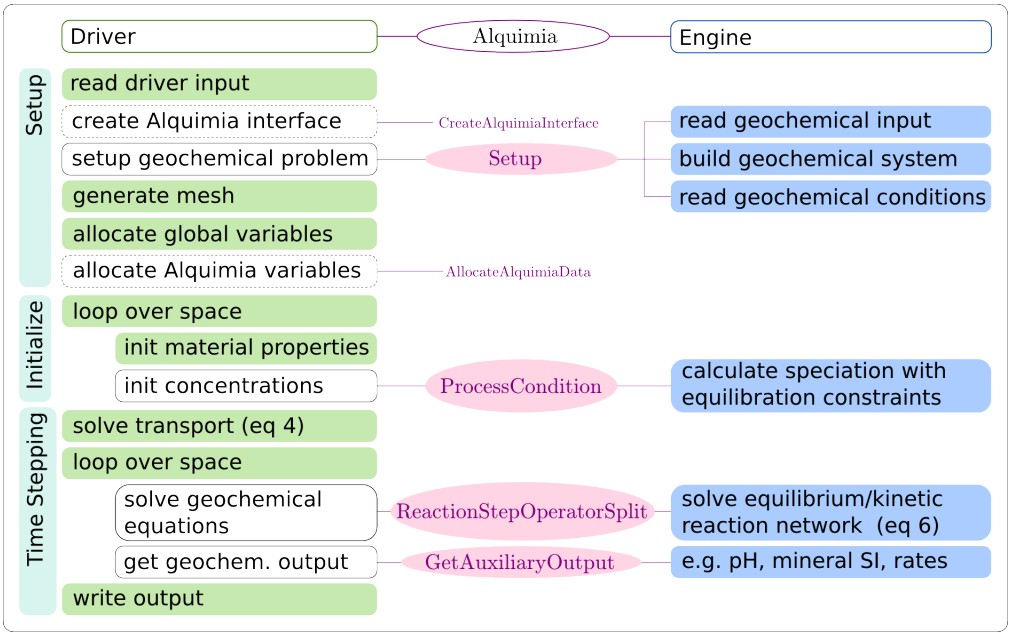

**Figure 1.** Example of a workflow in a code coupling a transport driver (left) and a geochemical engine (right) via Alquimia indicating the responsibilities for each the driver (green), Alquimia (pink) and engine (blue) codes. Calls to Alquimia functions in the driver code (unfilled boxes) replace development of native capabilities. Calls to Alquimia's utility library are indicated with dashed lines in the bounding box. The prefix Alquimia is omitted from Alquimia functions for clarity (i.e. *AlquimiaSetup, AlquimiaProcessCondition, AlquimiaReactionStepOperatorSplit, AlquimiaGetAuxiliaryOutput*).

While the transport step requires the solution over the entire domain, the geochemical equations can be solved independently within each cell. By separating the problem into two steps, operator splitting allows for solving the transport problem with linear solvers, confining the nonlinearity to the geochemical problem within a single cell. The time step size, however, is limited by the Courant criterion to avoid operator splitting error (Steefel and MacQuarrie, 1996).

## 3 Interface description

### 3.1 Functions

The Alquimia interface is designed to act as a generic, intermediary layer between a code that solves Eqs. (5-7) and a code that solves Eqs. (8). We will refer to the former code as the *driver* and the latter as the *engine* (Fig. 1).

The *driver* is the code that drives the simulation, handles the spatial description of the problem, including the meshing and spatial discretization, and solves Eqs. (5-7). It is responsible for managing global variable storage, including reading spatially and/or temporally varying material properties, looping through space, managing time stepping, unpacking and moving data from the mesh dependent storage into data transfer containers, and input/output (I/O) operations.





The geochemical *engine* defines the geochemical problem and solves Eq. 8 at each point in space independently. It is responsible for reading the geochemical reaction data, managing the geochemical system (e.g. reading the thermodynamic database), reading and processing the initial and boundary solutions (e.g. equilibrating the solution with specific minerals, or a pH value), and providing access to geochemical data for output (pH, mineral saturation indices, reaction rates).

The Alquimia interface itself handles the engine-dependent implementation for setting up, processing the speciation con-
straints, performing the reaction step solution, and shutting down. Alquimia does not do any geochemical calculations, but does perform any unit conversions required by each engine. At the start of each operation, it unpacks data from the Alquimia data transfer containers provided by the engine, and packages them into the correct format for that engine. Then, calls to engine subroutines are made to perform the appropriate calculations. At the end of the operation, it packages the results back into the Alquimia containers for use by the driver.

## 130    3.2    Software

Alquimia has two parts: (1) an engine-independent application programming interface (API) consisting of function call signatures, data structures, and constants (Table 1) and (2) an optional utility library.

The API works by enforcing a signature for geochemical subroutines using a single-cell model. That is, the calls to the geochemical solver are carried out for a single element of a given spatial discretization of a driver, and thus are made within a
loop over space.

The main calls in the workflow include *AlquimiaSetup*, *AlquimiaProcessCondition*, and *AlquimiaReactionStepOperator-Split*. *AlquimiaSetup* initializes the engine by reading the geochemical engine's input file and database. This builds the geochemical system, sets the values of the reaction parameters, and generates a list of aqueous solutions with appropriate equilibration constraints. *AlquimiaProcessCondition* performs the speciation calculations on this list of aqueous solutions to obtain
initial and boundary concentrations as needed by the engine, solving a steady-state form of Eqs. (8-5), with additional constraints such as fixed species concentration or pH values, and charge balance or mineral equilibrium. *AlquimiaReactionStep-OperatorSplit* performs the solution of the geochemical problem (Eq. 8-5).

Variables that may change in the engine with each call to reaction time-stepping are part of Alquimia's *AlquimiaState* data structure, including porosity, fluid density and pressure and the total concentrations ($\Psi_i$ , $\Psi_{m,i}$ , and $\Psi_{s,i}$), the reactive surface
areas of the minerals, the surface site density for sorption and the cation exchange capacity. Variables that do not change in the engine are part of the *AlquimiaProperties* data structure, including water saturation and cell volume and miscellaneous reaction parameters (sorption constants, Freundlich sorption exponents, Langmuir sorption coefficients, mineral rate constants, and kinetic reaction rate constants). The units used for each set of variables are required by the interface as outlined in the documentation. This implies that the driver must supply in the values in these units but it is the responsibility of the Alquimia
interface implementation of each engine to perform the necessary conversions to the engine's internal units.

Other structures contain information about the available functionality in the engine (*AlquimiaFunctionality*) (e.g. is porosity updated by the engine?), the status of the geochemistry engine after the last operation (*AlquimiaEngineStatus*) (e.g. did the solution converge for the time step?), data such as names of all species (*AlquimiaProblemMetaData*), and data for output





**Table 1.** Summary of Alquimia's API (data transfer containers, functions, constants) and C utilities library

| Data Transfer Containers | |
| --- | --- |
| struct: AlquimiaVectors | struct: AlquimiaEngineFunctionality |
| void pointer: engine_state | struct: AlquimiaProblemMetaData |
| struct: AlquimiaSizes | struct: AlquimiaAuxiliaryOutputData |
| struct: AlquimiaState | struct: AlquimiaGeochemicalCondition |
| struct: AlquimiaProperties | struct: AlquimiaAqueousConstraint |
| struct: AlquimiaAuxiliaryData | struct: AlquimiaMineralConstraint |
| struct: AlquimiaEngineStatus | |
| **Functions** | |
| void AlquimiaSetup | void AlquimiaProcessCondition |
| void AlquimiaShutdown | void AlquimiaReactionStepOperatorSplit |
| void AlquimiaGetEngineMetaData | void AlquimiaGetAuxiliaryOutput |
| **Constants** | |
| error codes | |
| string lengths | |
| strings | |
| **C Utilities Library** | |
| struct: AlquimiaInterface | void AllocateAlquimiaXXX |
| void CreateAlquimiaInterface | void FreeAlquimiaXXX |
| struct: AlquimiaData | void PrintAlquimiaXXX |
| void AllocateAlquimiaData | |

purposes (*AlquimiaAuxiliaryOutputData*). Another structure (*AlquimiaAuxiliaryData*) is used to store data necessary for the engine, the type of which is known but on which the driver should not do any operation. These data include for example the initial guesses for the next nonlinear solution, and thus the driver must return them on the next call and write them to checkpoint files. Last, a standalone pointer variable contains all the persistent internal state data for the chemistry engine that is not mesh dependent and can be reinitialized from the input file upon restart (*AlquimiaEngineState*).

Generally, the parameter values of the geochemical model are set by the engine. This simplifies code development on the driver side as the engines already have facilities to read in these parameters. However, there are cases where finer-grained control of certain parameter values by the driver. For example, in parameter estimation simulations or inverse problems that involve varying these values, it may be easier to control model parameters from the driver rather than having to write scripts that change the values in engine input files. For this purpose, Alquimia provides the ability to control a limited set of reaction parameters. These are part of the *AlquimiaProperties* data structure and include the linear sorption constant, the Freundlich





sorption exponent, the Langmuir sorption coefficient, the mineral dissolution/precipitation kinetic rate constant, and the aqueous kinetic reaction rate constant. A single flag controls this behavior. When turned on ("hands-on" mode), Alquimia will use the parameter values coming from the driver for each cell at every time step to populate the appropriate data holders in the engine. Otherwise ("hands-off" mode), the default behavior is to let the engine control these parameters. While enabling this option gives the driver more control over some engine parameters, this approach is involved in terms of coding and must be

used with caution, especially because it may expand the capabilities of the engines. For example, one can specify a different value of the rate constants in each grid cell, which in general is not an option available in most geochemical models.

The second part of the Alquimia library is a C utility library that contains reusable code for common tasks such as allocating memory, printing data and other miscellaneous auxiliary tasks. These are optionally used in driver codes to facilitate implementation of these common tasks.

For wide compatibility with mixed language programming, Alquimia is implemented in the C language as it offers the most flexibility to mix with other languages, including C++, Python, or Fortran. Examples of this flexibility are given in Secs. 4 and 5. The two engines currently available in Alquimia are implemented in Fortran (Sec. 4.1).

### 3.3   Development practices

Alquimia adheres to best practices set forth by the Extreme-scale Scientific Software Development Kit (xSDK) (Bartlett et al.,
2017). Among other things, this means that Alquimia uses a CMake-based build system, provides a comprehensive test suite, provides a documented, reliable way to contact the development team, and has an accessible repository.

The required parts of the Alquimia are compiled using CMake-based build system into libalquimia_c.a and libalquimia_fortran.a. Semantic versioning is used for its public API. The source code and tagged releases are hosted in a GitHub repository, publicly available via a three-clause BSD licence. A test suite is provided that is based on two simple driver codes: one for batch
geochemistry calculations and another for coupled reactive transport calculations. Both the build system and the test suite are included in the automated continuous integration framework available from GitHub Actions (GitHub, 2024), which is triggered by pull requests and used as condition for their approval. Alquimia is documented using restructured text files included in the source distribution and may be built and exported to different formats using Python's Sphinx package (Sphinx, 2024).

### 4   Implementation and use

The implementation and use of Alquimia are illustrated here by describing selected examples of the two tasks that Alquimia separates for engine and driver codes. One is the implementation of the geochemical calls in driver codes (e.g. left column in Fig. 1). This implementation is independent of the engines available; it only depends on Alquimia's data structures and call signatures. The second task is the implementation of the engine-specific function calls in Alquimia for a given engine (e.g. right column in Fig. 1). This also includes the necessary transfer of data between Alquimia data transfer containers and the engine's
internal data structures. This implementation is independent of the use any driver makes of Alquimia and does not need to be repeated every time the interface is implemented in a new driver code. That is, if a new engine is added, no changes are needed





in any driver that uses Alquimia to make use of these new engine's capabilities. This allows for performing simulations using the same driver replacing the geochemical engine. We use this feature to compare results from simulations performed using different driver-engine combinations of the codes presented in what follows.

## 4.1 Geochemical engines

The widely-used open-source codes PFLOTRAN and CrunchFlow have been implemented as engines in Alquimia. PFLO-TRAN is an open-source, massively parallel multiscale and multiphysics code for subsurface multiphase flow, reactive transport, geomechanics and geophysics applications (Hammond et al., 2014; Lichtner et al., 2015; Jaysaval et al., 2023). Crunch-Flow is an open-source software package for simulating reactive transport (Steefel et al., 2015). Although both codes also solve for flow and transport processes and are known for implementing the global implicit approach, they also give the user the possibility of running reactive transport simulations in operator splitting mode, e.g. Steefel and MacQuarrie (1996). This facilitated the isolation of the geochemical capabilities of these codes for implementation in Alquimia. Although this applies to *AlquimiaSetup*, *AlquimiaProcessCondition*, and *AlquimiaReactionStepOperatorSplit*, we focus here on the latter function for brevity to exemplify the steps to implement engine capabilities in Alquimia.

The implementation of *AlquimiaReactionStepOperatorSplit* for both engines follows essentially the same steps. These include copying the data from the transfer containers, passing the initial guesses to the appropriate variables, performing the iterative nonlinear solve of the geochemical problem, and upon checking for convergence updating mineral concentrations (Code blocks 1-2). However, the details of the implementation differ somewhat owing to the differences between the two codes.

**Code Block 1.** Selected sections of Alquima's implementation of the PFLOTRAN operator splitting step in *AlquimiaReactionStepOperator-Split*. Lines starting with "!..." indicate portions of the code omitted for brevity. Some sections have been edited for legibility. The reader is directed to the actual code for full details.

```
1:  subroutine ReactionStepOperatorSplit(pft_engine_state, &
2:      delta_t, properties, state, aux_data, status)
3:    ! ... include and "use module" statements, and variable declarations
4:    ! ... assigning of the target of a C pointer to the Fortran engine_state pointer, check for
        integrity
5:    reaction => engine_state%reaction
6:    call CopyAlquimiaToAuxVars(copy_auxdata, engine_state%hands_off, &
7:        state, aux_data, properties, &
8:        reaction, engine_state%global_auxvar, &
9:        engine_state%material_auxvar, engine_state%rt_auxvar)
10:   ! copy free ion primaries into initial guess array
11:   allocate(guess(reaction%ncomp))
12:   do i = 1, reaction%naqcomp
13:     guess(i) = engine_state%rt_auxvar%pri_molal(i)
14:   enddo
15:   do i = 1, reaction%immobile%nimmobile
```



```
16:         guess(i+reaction%offset_immobile) = engine_state%rt_auxvar%immobile(i)
17:     enddo
18:     ! perform batch calculation
19:     vol_frac_prim = 1.0
20:     engine_state%option%tran_dt = delta_t
21:     call RReact(guess, engine_state%rt_auxvar, engine_state%global_auxvar, &
22:         engine_state%material_auxvar, num_newton_iterations, &
23:         reaction, natural_id, engine_state%option, &
24:         PETSC_FALSE, PETSC_FALSE, ierror)
25:     deallocate(guess)
26:     ! check for errors, copy solution back to alquimia data struct
27:     if (ierror /= 1) then
28:         call RUpdateKineticState(engine_state%rt_auxvar, engine_state%global_auxvar, &
29:             engine_state%material_auxvar, engine_state%reaction, engine_state%option)
30:         call CopyAuxVarsToAlquimia( &
31:             engine_state%reaction, &
32:             engine_state%global_auxvar, &
33:             engine_state%rt_auxvar, &
34:             porosity, &
35:             state, aux_data)
36:     else
37:         ! ...
38:     endif
39: end subroutine ReactionStepOperatorSplit
```

PFLOTRAN uses an object-oriented programming model introduced in a code refactoring (Hammond, 2022), while Crunch-Flow uses a legacy procedural modular programming. These two different approaches require different ways of storing and manipulating engine data within Alquimia.

In PFLOTRAN, most data structures needed to describe the geochemical problem are passed as arguments and thus must be included as part of the *AlquimiaEngineState* in Alquimia. This include those that contain geochemical reaction data such as stoichiometric coefficients, *engine_state%reaction*, global variables such as aqueous saturation, *engine_state%global_auxvar*, reactive transport variables such as total concentrations, *engine_state%rt_auxvar*, or material properties such as porosity, *engine_state%material_auxvar*. As shown in Code Block 1, handling this within Alquimia is straigthforward using *AlquimiaEngineState* and also makes it for easy-to-maintain code. When new variables are added in PFLOTRAN the Alquimia interface does not need to change; only if the capabilities of Alquimia are expanded and there are new variables that need to be passed explicitly, the interface require modification. These changes would mostly be limited to *CopyAlquimiaToAuxVars* and *CopyAuxVarsToAlquimia*, which are helper subroutines that copy data from Alquimia transfer containers to the engine's variables and back.

CrunchFlow relies on global variables declared in modules, which are dynamically allocated upon initialization from the inputs. This requires that these modules are included in the Alquimia interface. Examples include aqueous saturation (*satliq*)





from the transport module, porosity (*por*) from the medium properties module, or total concentrations (*sn*) from the concentration module (see *CopyAlquimiaToAuxVars* subroutine in *crunch_alquimia_interface.F90*). By contrast, CrunchFlow passes the dimensions of the geochemical problem from high-level subroutines to low-level subroutines, thus the *AlquimiaEngineState*

data structure is used to store them (e.g. *engine_state%ncomp*, *engine_state%nspec*, *engine_state%nkin*, among other geochemical sizes). While Alquimia's approach to engine data sometimes introduces more detail into code, its flexibility allows Alquimia to accommodate the needs of very different engines.

**Code Block 2.** Selected sections of Alquimia implementation of the CrunchFlow operator splitting step in *AlquimiaReactionStepOperatorSplit*. Lines starting with "!..." indicate portions of the code omitted for brevity. Some sections have been edited for brevity. The reader is directed to the actual code for full details.

```
1: subroutine ReactionStepOperatorSplit(cf_engine_state, &
2:   delta_t, properties, state, aux_data, status)
3:   ! ... "use module" statements and variable declarations
4:   ! ... assigning of the target of a C pointer to the Fortran engine_state pointer, check for
        integrity
5:   call CopyAlquimiaToAuxVars(copy_auxdata,engine_state%hands_off,state,aux_data,properties,&
6:    engine_state%ncomp,nspec,engine_state%nkin,engine_state%nrct,engine_state%ngas,&
7:    engine_state%nexchange,engine_state%nsurf,engine_state%ndecay,engine_state%npot,&
8:    engine_state%nretard)
9:   ! ...
10:  delt = delta_t / secyr
11:  newtmax = 0
12:  jx=1; jy=1; jy=1
13:  CALL keqcalc2(ncomp,nrct,nspec,ngas,nsurf_sec,jx,jy,jz)
14:  IF (igamma == 3) THEN
15:    IF (Duan) THEN
16:      CALL gamma_co2(ncomp,nspec,ngas,jx,jy,jz)
17:    ELSE
18:      CALL gamma(ncomp,nspec,jx,jy,jz)
19:    END IF
20:  END IF
21:  CALL AqueousToBulkConvert(jx,jy,jz,AqueousToBulk)
22:  CALL os3d_newton(engine_state%ncomp,engine_state%nspec,engine_state%nkin,nrct,engine_state%ngas,&
23:    engine_state%ikin,engine_state%nexchange,engine_state%exch_sec,engine_state%nsurf,&
24:    engine_state%nsurf_sec,engine_state%npot,engine_state%ndecay,engine_state%neqn,&
25:    engine_state%igamma,delt,engine_state%corrmax,engine_state%jx,engine_state%engine_state%jy,&
26:    engine_state%jz,iterat,icvg,engine_state%nx,engine_state%ny,engine_state%nz,time,AqueousToBulk)
27:  CALL SpeciesLocal(ncomp,nspec,jx,jy,jz)
28:  CALL totconc(ncomp,nspec,jx,jy,jz)
29:  ! ...
30:  if (icvg == 1) then
31:  ! not converged, do nothing
32:    ! ... recover appropriate initial guesses
```




```
33:   else
34:     CALL mineral_update(nx,ny,nz,nrct,delt,dtnewest,ineg,jpor,deltmin)
35:     ! ...
36:   end if
37:   ! ... update variables holding previous time step solutions with new
38:   ! send state variables back to Alquimia state with solution of this solve
39:   call CopyAuxVarsToAlquimia(engine_state%ncomp,engine_state%nspec,engine_state%nkin,&
40:     engine_state%nrct,engine_state%ngas,engine_state%nexchange,engine_state%nsurf,&
41:     engine_state%ndecay,engine_state%npot,engine_state%nretard,state,aux_data)
42:   return
43: end subroutine ReactionStepOperatorSplit
```

## 4.2  Drivers

The widely-used open-source codes Amanzi and ParFlow use Alquimia to implement geochemical capabilities. Amanzi is a multi-process high-performance-computing simulator that provides a flexible and extensible simulation capability. ParFlow is a parallel, integrated hydrology model that simulates spatially distributed surface and subsurface flow, as well as land surface processes including evapotranspiration and snow. The implementation of Alquimia in these codes responded to the particular needs and capabilities in each case.

**Code Block 3.** Amanzi implementation of Alquima's *AlquimiaReactionStepOperatorSplit*. Lines starting with "// ..." indicate portions of the code omitted for brevity.

```
1: bool
2: ChemistryEngine::Advance(const double delta_time,
3:                          const AlquimiaProperties& mat_props,
4:                          AlquimiaState& chem_state,
5:                          AlquimiaAuxiliaryData& aux_data,
6:                          AlquimiaAuxiliaryOutputData& aux_output,
7:                          int& num_iterations)
8: {
9:   // ...
10:
11:   // Advance the chemical reaction all operator-split-like.
12:   chem_.ReactionStepOperatorSplit(&engine_state_,
13:                                   delta_time,
14:                                   &(const_cast<AlquimiaProperties&>(mat_props)),
15:                                   &chem_state,
16:                                   &aux_data,
17:                                   &chem_status_);
18:   // ...
19:
20:   // Retrieve auxiliary output.
21:   chem_.GetAuxiliaryOutput(&engine_state_,
```





```
22:                        &(const_cast<AlquimiaProperties&>(mat_props)),
23:                        &chem_state,
24:                        &aux_data,
25:                        &aux_output,
26:                        &chem_status_);
27:
28:    // Did we succeed?
29:    if (chem_status_.error != kAlquimiaNoError) return false;
30:
31:    // Did we converge?
32:    if (!chem_status_.converged) return false;
33:
34:    // Write down the (maximum) number of Newton iterations.
35:    num_iterations = chem_status_.num_newton_iterations;
36:    return true;
37: }
```

In Amanzi, unstructured-mesh and structured-mesh discretizations are available in a single C++ code base and Alquimia was the right solution for a unified geochemical interface that worked for the different data structures holding the state variables for both meshes. (In this work we use the labels Amanzi-U and Amanzi-S for convenience to distinguish between unstructured and structured capabilities, respectively.) *ChemistryEngine::Advance* is the C++ function that provides a unified access within Amanzi to Alquimia's *AlquimiaReactionStepOperatorSplit* C function (Code Block 3). After calling *AlquimiaReactionStepOperatorSplit*, it obtains the auxiliary output variables. This is by choice here; there is no requirement to do so every time step but in Amanzi, it is done so that when requested by the user, certain output variables can be written to the output file. This function is called for each cell of the discretization, thus the driver is responsible for handling how their data structures are accessed for each cell.

The structured-mesh capabilities rely on block-structured adaptive mesh refinement (AMR) from the BoxLib library (Dubey et al., 2014), build upon Fortran Array Boxes (*FArrayBox*). The function that solves the geochemical problem is written for this *FArrayBox*. Using the available box iterators, the codes iterates for each grid cell in the box, performing three operations (Code Block 4). First, the Alquimia transfer containers are updated using the structured variables, then the *ChemistryEngine::Advance*) function (described above) is called with these updated data structures, solving the geochemical problem, and last the new solution is passed back to the (*FArrayBox*) structures for the next transport step.

**Code Block 4.** Amanzi-S call to advance chemistry. Lines starting with "// ..." indicate portions of the code omitted for brevity.

```
1: void
2: AlquimiaHelper_Structured::Advance(const FArrayBox& aqueous_saturation,    int sSat,
3:                                    const FArrayBox& aqueous_pressure,      int sPress,
4:                                    const FArrayBox& porosity,              int sPhi,
5:                                    const FArrayBox& volume,                int sVol,
6:                                    FArrayBox&       primary_species_mobile, int sPrimMob,
7:                                    FArrayBox&       fcnCnt,                 int sFunc,
```





```
8:                                    FArrayBox&       aux_data, Real water_density, Real temperature,
9:                                    const Box& box, Real dt, int chem_verbose)
10: {
11:     // ...
12:     Box thread_box(box);
13:     thread_box.setSmall(BL_SPACEDIM-1,tli);
14:     thread_box.setBig(BL_SPACEDIM-1,tli);
15:
16:     for (IntVect iv=thread_box.smallEnd(), End=thread_box.bigEnd(); iv<=End; thread_box.next(iv)) {
17:
18:       BL_to_Alquimia(aqueous_saturation,      sSat,
19:                      aqueous_pressure,        sPress,
20:                      porosity,                sPhi,
21:                      volume,                  sVol,
22:                      primary_species_mobile,  sPrimMob,
23:                      aux_data, iv, water_density, temperature,
24:                      alquimia_properties[threadid],
25:                      alquimia_state[threadid],
26:                      alquimia_aux_in[threadid],
27:                      alquimia_aux_out[threadid]);
28:
29:       // ...
30:
31:       int newton_iters;
32:       engine->Advance(dt,alquimia_properties[threadid],alquimia_state[threadid],
33:                       alquimia_aux_in[threadid],alquimia_aux_out[threadid],newton_iters);
34:
35:       // ...
36:
37:       Alquimia_to_BL(primary_species_mobile,  sPrimMob,
38:                      aux_data, iv,
39:                      alquimia_properties[threadid],
40:                      alquimia_state[threadid],
41:                      alquimia_aux_in[threadid],
42:                      alquimia_aux_out[threadid]);
43:
44:       fcnCnt(iv,sFunc) = newton_iters;
45:     }
46:   }
47: }
```

The three operations are also present in the unstructured counterpart of the *Advance* function (Code Block 5). The copy operations from and to Alquimia data structures are different than those in the structured function and thus cannot be re-used.





However, the call to the chemistry engine *Advance* is the same as in the structured function. This exemplifies how the single-cell model adopted in Alquimia offers significant flexibility in enabling a broad range of discretizations.

**Code Block 5.** Amanzi-U call to advance chemistry

```
 1: int
 2: Alquimia_PK::AdvanceSingleCell(double dt,
 3:                                Teuchos::RCP<Epetra_MultiVector>& aqueous_components,
 4:                                int cell)
 5: {
 6:   // Copy the state and property information from Amanzi's state within
 7:   // this cell to Alquimia.
 8:   CopyToAlquimia(
 9:     cell, aqueous_components, alq_mat_props_, alq_state_, alq_aux_data_, Tags::DEFAULT);
10:
11:   int num_iterations = 0;
12:   if (alq_mat_props_.saturation > saturation_tolerance_) {
13:     bool success = chem_engine_->Advance(
14:       dt, alq_mat_props_, alq_state_, alq_aux_data_, alq_aux_output_, num_iterations);
15:     if (not success) {
16:       if (vo_->os_OK(Teuchos::VERB_MEDIUM)) {
17:         Teuchos::OSTab tab = vo_->getOSTab();
18:         *vo_->os() << "no convergence in cell: " << mesh_->cell_map(false).GID(cell) << std::endl;
19:       }
20:       return -1;
21:     }
22:   }
23:
24:   // Move the information back into Amanzi's state, updating the given total concentration vector.
25:   CopyAlquimiaStateToAmanzi(
26:     cell, alq_mat_props_, alq_state_, alq_aux_data_, alq_aux_output_, aqueous_components);
27:
28:   return num_iterations;
29: }
```

In ParFlow, earlier work entailed coupling solute transport in the subsurface to a subset of the geochemical capabilities in CrunchFlow via a custom interface (Beisman et al., 2015). In that case, the same C-to-Fortran macros used for coupling Parflow

with the CLM land model (Maxwell and Miller, 2005) were applied. This earlier work served as the basis for the implementation of Alquimia in the code, although the availability of Alquimia's C interface here enabled a more seamless coupling, without the need of C to Fortran macros. For example, the *AlquimiaReactionStepOperatorSplit* C function is called directly within a loop inside block structured discretization (Code Block 6). The function in Code Block 6 shows how the coupling utilizes extensively the C utility library. ParFlow calls *CopyAlquimiaState*, *CopyAlquimiaProperties*, and *CopyAlquimiaAux-*

*iliaryData* are used to store and retrieve data before and after the call to *AlquimiaReactionStepOperatorSplit* in case of non



convergence of the solution (Code Block 6). This is in contrast to Amanzi, where similar functions are used but they are written as new C++ code in Amanzi itself and encapsulated in *BL_to_Alquimia* and *Alquimia_to_BL* for the structured capabilities and *CopyToAlquimia* and *CopyAlquimiaStateToAmanzi* for the unstructured.

**Code Block 6.** Parflow call to Alquima. Lines starting with "// ..." indicate portions of the code omitted for brevity.

```
1:  void AdvanceChemistry(ProblemData *problem_data, AlquimiaDataPF *alquimia_data, Vector **
        concentrations, Vector *saturation, double dt, double t, int *any_file_dumped, int dump_files,
        int file_number, char *file_prefix)
2:  {
3:    // ...
4:    // copy the transported primary mobile concentrations to alquimia
5:    AdvectedPrimaryToChem(alquimia_data->chem_state, &alquimia_data->chem_sizes, concentrations,
        problem_data);
6:
7:    // solve chemistry cell-by-cell
8:    ForSubgridI(is, subgrids)
9:    {
10:     subgrid = SubgridArraySubgrid(subgrids, is);
11:
12:     // ...
13:     por_sub = VectorSubvector(ProblemDataPorosity(problem_data), is);
14:     por = SubvectorData(por_sub);
15:     sat_sub = VectorSubvector(saturation, is);
16:     sat = SubvectorData(sat_sub);
17:
18:     GrGeomInLoop(i, j, k, gr_domain, r, ix, iy, iz, nx, ny, nz,
19:       {
20:         por_index = SubvectorEltIndex(por_sub, i, j, k);
21:         sat_index = SubvectorEltIndex(sat_sub, i, j, k);
22:
23:         chem_index = (i-ix) + (j-iy) * nx + (k-iz) * nx * ny;
24:
25:         alquimia_data->chem_properties[chem_index].volume = vol;
26:         alquimia_data->chem_properties[chem_index].saturation = sat[sat_index];
27:
28:         // Set the thermodynamic state.
29:         alquimia_data->chem_state[chem_index].water_density = water_density;
30:         alquimia_data->chem_state[chem_index].temperature = 25.0;
31:         alquimia_data->chem_state[chem_index].porosity = por[por_index];
32:         alquimia_data->chem_state[chem_index].aqueous_pressure = aqueous_pressure;
33:
34:         //copy pre-solution state and aux data to temp containers in case non-convergence is an issue
35:         CopyAlquimiaState(&alquimia_data->chem_state[chem_index], &alquimia_data->chem_state_temp);
```



```
36:        CopyAlquimiaAuxiliaryData(&alquimia_data->chem_aux_data[chem_index], &alquimia_data->
       chem_aux_data_temp);
37:        CopyAlquimiaProperties(&alquimia_data->chem_properties[chem_index], &alquimia_data->
       chem_properties_temp);
38:
39:        // Solve the geochemical system
40:        alquimia_data->chem.ReactionStepOperatorSplit(&alquimia_data->chem_engine,
41:                                         dt_seconds, &alquimia_data->chem_properties[chem_index],
42:                                         &alquimia_data->chem_state[chem_index],
43:                                         &alquimia_data->chem_aux_data[chem_index],
44:                                         &alquimia_data->chem_status);
45:        if (alquimia_data->chem_status.error != 0)
46:        {
47:          amps_Printf("ReactionStepOperatorSplit() error: %s\n",
48:                      alquimia_data->chem_status.message);
49:          PARFLOW_ERROR("Geochemical engine error, exiting simulation.\n");
50:        }
51:
52:        if (!(alquimia_data->chem_status.converged))
53:        {
54:          // ...
55:        }
56:
57:        alquimia_data->chem.GetAuxiliaryOutput(&alquimia_data->chem_engine,
58:                                         &alquimia_data->chem_properties[chem_index],
59:                                         &alquimia_data->chem_state[chem_index],
60:                                         &alquimia_data->chem_aux_data[chem_index],
61:                                         &alquimia_data->chem_aux_output[chem_index],
62:                                         &alquimia_data->chem_status);
63:        if (alquimia_data->chem_status.error != 0)
64:        {
65:          amps_Printf("GetAuxiliaryOutput() auxiliary output fetch failed: %s\n",
66:                      alquimia_data->chem_status.message);
67:          PARFLOW_ERROR("Geochemical engine error, exiting simulation.\n");
68:        }
69:    });
70:  }
71:
72:  // copy solved primary concentrations back to PF
73:  ReactedPrimaryToPF(alquimia_data->chem_state, &alquimia_data->chem_sizes, concentrations,
       problem_data);
74:
75:  // ...
76: }
```





### 4.3 Multi-way comparison

We build on the tests in the Alquimia test suite to develop a set of one-dimensional simulations of reactive transport with Amanzi and ParFlow. The test suite (Sec. 3.3) ensures the correct functioning in the simulation of specific reaction types separately: mineral dissolution-precipitation, aqueous kinetics, ion exchange, surface complexation, and isotherm-based sorption in batch and reactive transport scenarios. In addition, a non-reactive tracer simulation is used to identify how the different discretization schemes affect the differences in the results but also to rule out numerical issues by the data transfer steps involved in using Alquimia.

The simulations are simple with regard to the transport processes and the spatial distribution of properties in this domain. The domain is one-dimensional, 100 meters in length and discretized with 100 cells, with a porosity of 0.25. A uniform flow rate is applied along the domain in fully-saturated conditions such as the infiltrating front is half-way through the domain in the 50-year simulations. Diffusive-dispersive processes are not considered. In the tests with heterogeneous reactions, the solution initially in the domain is in equilibrium with the mineral or surface. A solution with a distinct composition infiltrates from the left boundary, displacing the initial solution and driving the geochemical reactions considered in each case (except for the non-reactive tracer.)



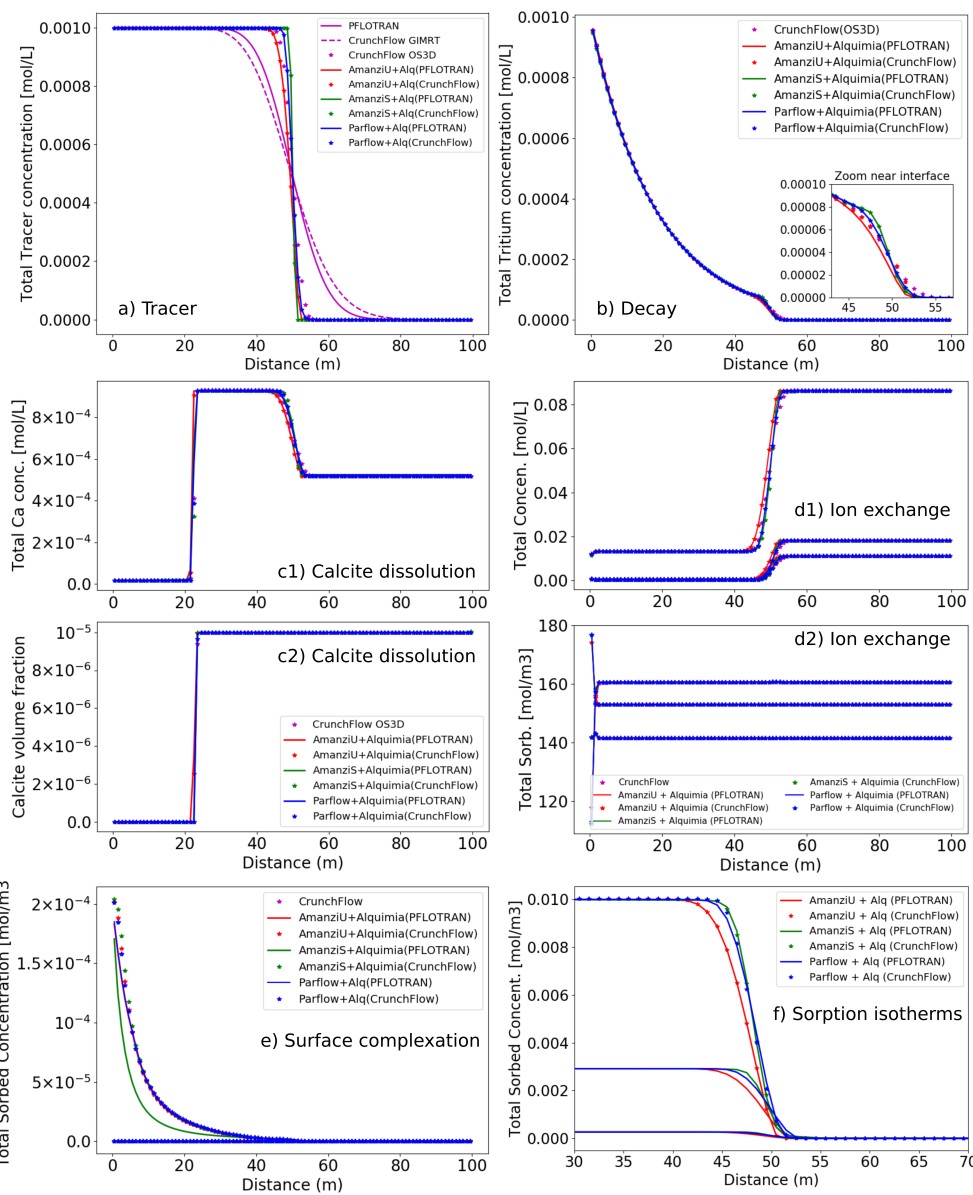

**Figure 2.** Selected results from six 1D reactive transport simulations that consider common geochemical reactions separately: (a) non-reactive tracer, (b) tritium decay, (c) calcite dissolution, (d) ion exchange, (e) surface complexation and (f) isotherm-based sorption. Each simulation was performed with different code combinations Amanzi (-S and -U) and Parflow were used as driver codes, with PFLOTRAN and CrunchFlow as engine codes. Additionally, PFLOTRAN and CrunchFlow were also used as reactive transport simulators, where GIMRT and OS3D refer to the global implicit and operator splitting capabilities of CrunchFlow. In global implicite mode, PFLOTRAN and CrunchFlow solve transport implicitly resulting in diffuse solutions, which are omitted in the figure. The Langmuir and Freundlich sorption isotherms presented for CrunchFlow as they are not directly available in CrunchFlow, although Langmuir could be readily implemented with a surface complexation model and a single sorbing species. Standalone CrunchFlow does not output sorbed concentrations for Kd linear sorption.





Each problem is simulated eight times. This includes six times with different driver-engine combinations (two each with Amanzi-S, Amanzi-U, and ParFlow, using PFLOTRAN and CrunchFlow as engines). Two additional simulations are performed with PFLOTRAN and CrunchFlow as standalone codes using their own transport solvers. In all cases, the Courant number was kept to 1, as needed by the operator splitting approach.

Results from the non-reactive tracer show that the advective front is at 50 m at 50 years (Figure 2a). Spreading of the front can be attributed exclusively to numerical dispersion added by the numerical scheme employed in each case. Results from Amanzi-S show the sharpest advective front, consistent with the high-order methods in BoxLib (Dubey et al., 2014), especially when Courant number equals 1. CrunchFlow's third-order time-diminishing variation (TVD) scheme (Gupta et al., 1991) in operator splitting mode results in a numerical dispersion similar to that of Amanzi-U's explicit second order scheme. PFLOTRAN and CrunchFlow in global implicit mode use both implicit solvers that result in larger numerical dispersion.

The differences in the discretization schemes affect the results from the reactive transport simulations differently. For kinetically-controlled aqueous reactions such as first-order tritium radioactive decay, $r = -\lambda c$, differences arise at the leading edge of the infiltration front but disappear with time behind the front (Figure 2b). For fast or equilibrium heterogeneous reactions, two fronts may be present, one associated with the advective infiltration front and one with the heterogeneous reaction.

In the calcite simulation, dissolution is treated as a kinetic reaction with a transition-state-theory-type rate law:

$$R_i = kA(1 - Q/K_s) \tag{10}$$

where $k$ is the rate constant, $Q$ is the ion activity product, $K_s$ is the equilibrium constant of the reaction, and $A$ is the reactive surface area. The intrinsic rate of reaction ($kA$) is much faster than the rate of advective transport, which effectively results in equilibrium conditions. The incoming solution drives dissolution, depleting over time the initial mass of calcite. Where calcite is still present, the solution is in equilibrium with respect to calcite (between 22-100 m at 50 years, 2c). Where calcite has been depleted, the concentrations reflect the incoming solution (between 0-22 m at 50 years). The reactive front is thus sharp. All codes capture accurately the location of this sharp front, with only minor differences in the values in the grid cells near the front compared to the absolute variation in values in the front. Thus, this front is not as affected by the numerical dispersion discussed earlier. In contrast, the infiltration front at 50 m shares the same issues discussed previously with regard to the front spreading. The solution resulting from calcite dissolution upstream mixes with the initial solution owing to numerical dispersion (Figure 2c). In this example, the reaction reduce differences that may arise from the transport calculations such as those from discretization schemes. In reactive transport systems at steady-state or quasi-steady state conditions (such as this slow-moving calcite dissolution front), numerical dispersion is often less of a concern to practitioners in contrast to the small time steps required with the operator splitting approach.

Surface reactions (sorption isotherms, ion exchange, and surface complexation) are all treated as equilibrium reactions by the two engines available in Alquimia. Results for the associated test examples also show a very reasonable agreement between codes capturing the resulting fronts (Fig. 2). Sorption isotherms (linear-Kd, Freundlich and Langmuir) express relationships between aqueous and sorbed concentrations. As such, there is only the advective front present in the results, with the sorbed





concentrations tracking this front (Fig. 2f, compare with Fig. 2a), and the codes perform according to their performance in the conservative tracer simulations.

The ion exchange simulation shows two sets of fronts (Fig. 2d). The fronts are limited to very narrow bands near the inlet and at 50 m (not visible in Fig. 2d). The infiltrating solution is dilute in comparison to the initial solution but the proportions

of the solutes are different. Sorbed concentrations change accordingly, with $Mg^{2+}$ and $Ca^{2+}$ increasing relatively to initial and $Na^+$ decreasing. Surface complexation of $Zn^{2+}$ also shows good agreement between codes, where the greater $Zn^{2+}$ concentrations in the infiltrating solution result in increasing sorbed concentrations with time (Figure 2e).

## 5 Applications

This section presents examples of how Alquimia can be deployed within very different applications and purposes. Alquimia's

interface is well-defined (i.e. variables and parameters given and returned are unequivocally specified with their units, and functions are clearly described), and is flexible enough to allow for the peculiarities associated with each application, making it broadly applicable to the simulation of geochemical systems. From a software perspective, the interface must couple with codes written in different programming languages, and the code performance must not be hindered by the coupling. This section also documents issues associated with the development and how they were overcome in each case.

### 5.1 Pore-scale multiphase flow and reactive transport

In recent years, there has been an increase in the use of reactive transport models to simulate pore-scale processes, e.g. Molins et al. (2020, 2024b). A distinctive aspect of pore-scale models is that they represent explicitly the fluid and solid phases that make up porous media. From a geochemical perspective, this allows for consideration of mineral surface areas directly from the pore space geometry and thus for the accessibility of the reactive fluids to the mineral surfaces (Molins, 2015).

A recent area of interest is the coupling of pore-scale models for multiphase flow to reactive transport. In this direction, Li et al. (2022) developed a multiphase flow and reactive transport simulator building on capabilities of openFOAM (noa, 2022). OpenFOAM (for "Open-source Field Operation And Manipulation") is a very popular open-source platform written in C++ to develop computational models in fluid dynamics applications and beyond. For the application presented in Li et al. (2022), openFOAM provides the solver tools for multiphase flow and transport of solutes in the aqueous phase. However,

no geochemical packages are available to represent multicomponent aqueous speciation and mineral dissolution-precipitation reactions. CrunchFlow has been used previously in this role, including in pore-scale applications (Molins et al., 2012; Beisman et al., 2015; Zhang et al., 2022, 2024), and thus its use was a preferred choice.

Alquimia facilitated the development of the application. Because it is written in C, it was straightforward to use in Open-FOAM's C++-based code. It also provided enough flexibility to incorporate the pore-scale conceptualization of reactive pro-

640 cesses. Importantly for pore-scale applications, the reactive surface area ($A'$) in the mineral rate ($R'_i$) is in units of area. e.g., $m^2$. Mineral dissolution in the pore-scale model is simulated with a transition-state-theory-type rate law:

$$R'_i = kA'(1 - Q/K_s) \tag{11}$$





where $k$ is the rate constant, $Q$ is the ion activity product, $K_s$ is the equilibrium constant of the reaction, and $A'$ is the reactive surface area. Alquimia was originally designed with porous-media applications and it also assumes mass balance is performed

645  per unit volume. As such, it requires this surface area to be in units of specific area, namely $m^2/m^3 - bulk$ as the conservation equation 8 is written per unit volume. In other words, this implies that the surface area must be normalized by the volume of medium it occupies. In a discretized form, this volume is the volume of the grid cell $n$ ($V^n$). Hence, in the single-cell Alquimia model, the surface area is the area of the interface as computed by openFOAM normalized by the grid cell volume ($A^n = A'^n/V^n$). At the same time, the volumetric water content ($\theta$) on the left-hand side of Eq. 8 is set to be the water volume

in the cell normalized to the grid cell volume ($\theta^n = V_{aq}^n/V^n$). Hence, the function $f_k$ in Eq. 6, is in discretized form:

$$R_i^n = kA^n(1 - Q^n/K_s) \tag{12}$$

In a cell n where $A'^n$ is not zero (i.e. in contact with the solid phase) and where $V_{aq}^n$ is not zero (i.e. the aqueous phase is present), can be non-zero (i.e. dissolution or precipitation may take place).

OpenFOAM uses header files for setting up and initializing the problem at hand. This implies that initial values for primary

variables such as species concentrations in solute transport problems are given in these files. In multicomponent geochemical models, the value of the initial and boundary concentrations is generally not known before the simulation, rather, a set of constraints is given to obtain species concentrations. The specialized Alquimia *AlquimiaProcessCondition* function call is used for this purpose in Li et al. (2022). In a first step, the dimensions of the geochemical problem are given in an openFOAM header files, which then are used to allocate and initialize the concentration variables with dummy values. In a second step, the

Alquimia *AlquimiaSetup* function is called which provides the dimensions of the geochemical system, which must be checked for consistency with those in the header files, followed by a call to *AlquimiaProcessCondition* (Fig. 3) to initialize and set the boundary conditions of the pore-scale problem in OpenFOAM.





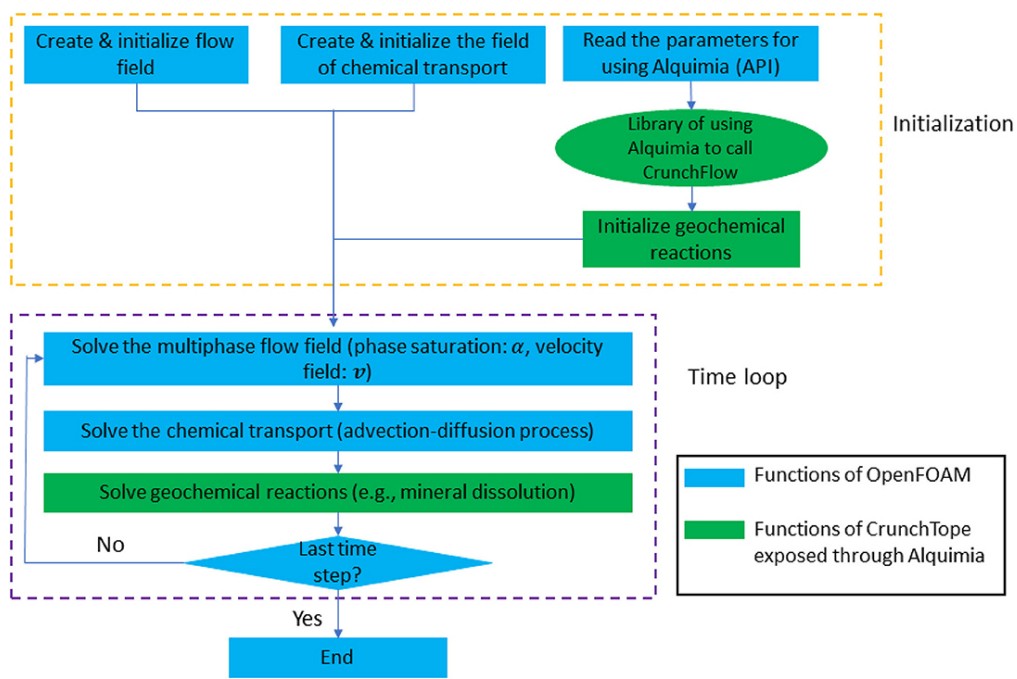

**Figure 3.** Diagram summarizing the computational workflow of CrunchFOAM, including initialization of the CrunchFlow geochemical problem and calls to its solver via Alquimia. Reprinted from Li et al. (2022)

## 5.2 Land surface processes

In the Earth System Model (ESM) context, the land surface represents the lower boundary of the atmosphere. It also acts
as host of biogeochemical cycles such as soil organic matter decomposition that influence the global carbon budget. Land
surface models (LSMs) couple the relevant processes in terrestrial environments in order to quantify key interactions including
greenhouse gas exchange fluxes between soil and atmosphere or the capacity of soils to sequester carbon. LSMs generally
use simplified representations of biogeochemical cycling that consider the role of carbon, nitrogen, phosphorus and water but
neglect other components, pH, and the aqueous reactions associated with it, including detailed terminal electron accepting
processes (Sulman et al., 2024). There is an opportunity to significantly improve LSMs by incorporating the broad range of
reactions that are available in general-purpose geochemical models. In particular, PFLOTRAN offers flexible options via its
Reaction Sandbox feature to readily implement custom reaction rate models (Hammond, 2022). LSMs are already complex
codes, in that they include many processes, and thus development often benefits from prototyping new process models, testing
them considering a reduced number of coupled processes or exploring scenarios systematically. This process typically also
benefits from using tools that automate processing and visualization of the results.

In this example application, Sulman and co-authors (Wang et al., 2024; Sulman et al., 2024) used Alquimia to incorporate
geochemical processes in land surface modeling. Rather than tackling a full implementation to an existing land surface model,
a two-step approach was used.



In a first step, a Python-based prototyping simulator was developed and applied to the simulations of methane processes

in Arctic soils. A Python interface implemented Alquimia's API functions and data structures as Python functions using the CFFI package (CFFI, 2023). Alquimia's C interface was used to couple it to this Python implementation. As a prototyping tool, the model consisted of a single-cell representation of the system and the main use was to systematically prescribe fluxes in the Python code directly. This allowed for setting up the domain in Python data structures while retaining the PFLOTRAN reaction network capabilities. Time-stepping was done in the Python code. The *hands-on* mode in Alquimia allowed setting

and updating reaction rate constants from the Python code in each time step. This is not necessarily possible with PFLOTRAN directly. This Python-Alquimia prototyping system was further applied to simulate one-dimensional soil column processes in coastal wetlands (Wang et al., 2024).

In a second step, Alquimia was used to couple PFLOTRAN to the existing Energy Exascale Earth System Model (E3SM) Land Model (ELM). This allowed for representing complex redox dynamics, aqueous and solid-phase chemistry, and pH dy-

690 namics in ELM (LaFond-Hudson and Sulman, 2023; Sulman et al., 2024). This is especially important in tidal wetlands, which are subject to both saltwater and freshwater inputs driven by tidal hydrological dynamics. Saltwater inputs inputs are associated with elevated sulfate concentrations that provide alternative terminal electron acceptors and reduce methane emissions in saltwater-affected wetlands. This work built on the prototype developed earlier but relied on the FORTRAN interface in Alquimia for convenience given that ELM is written in FORTRAN. Although PFLOTRAN is also in FORTRAN, from the

695 engine's perspective (ELM in this case), the implementation is independent of the language the engine is written. Alquimia initialization (*AlquimiaSetup*) and initial condition equilibration (*AlquimiaProcessCondition*) subroutines were added to the ELM initialization code, and the Alquimia time stepping subroutine was added to ELM (Figure 4).



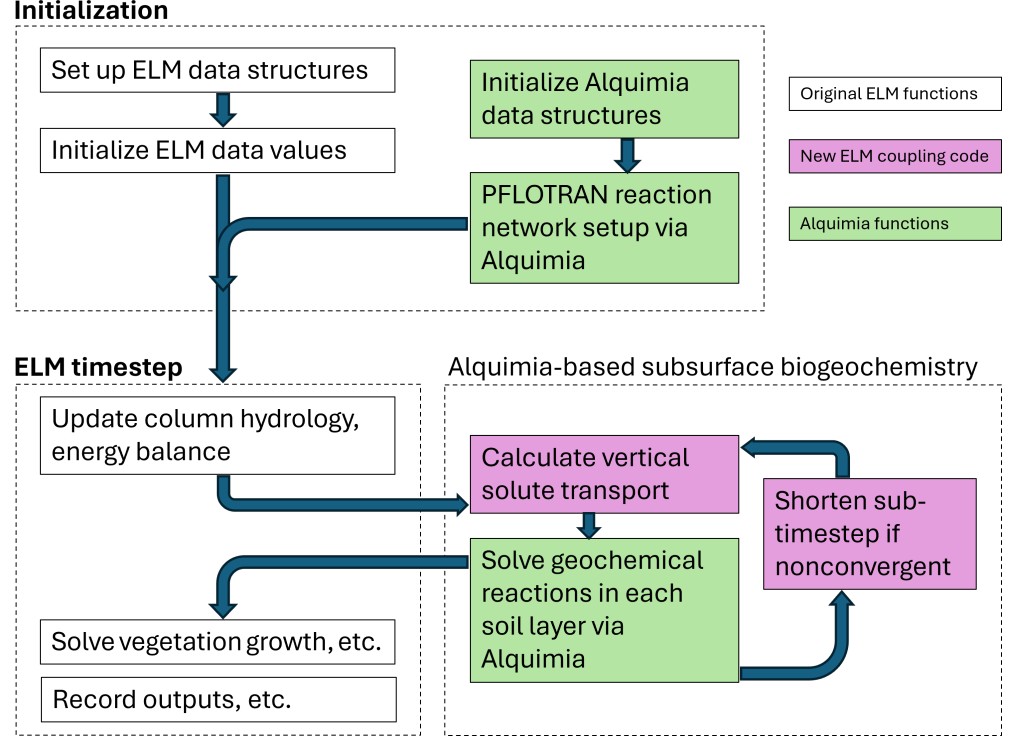

**Figure 4.** Diagram summarizing the computational workflow of ELM-PFLOTRAN, including initialization of the PLFOTRAN geochemical problem and calls to its solver via Alquimia.

ELM has its own description of biogeochemical processes, including consideration of carbon and nitrogen, which are present in multiple solid-state pools such as litter and soil organic matter, as well as aqueous soil nitrate and ammonium. In order to replace the ELM description of biogeochemical processes with that of PFLOTRAN, there has to be a one-to-one correspondence between ELM pools and the pools that are included in the PFLOTRAN reaction network used in the coupling (Tang et al., 2016). This is accomplished via the PFLOTRAN input file, along with soil organic matter decomposition kinetics defined using the PFLOTRAN Reaction Sandbox (Hammond, 2022). In this process, however, the Alquimia data structure interface was also augmented to treat solid-state SOM pools as immobile chemicals, allowing for transparent data transfer of the SOM pools from ELM to PFLOTRAN and back via the interface.

In the implementation, ELM decomposition processes were fully replaced by equivalent or modified calculations on the PFLOTRAN side and included the pools required by ELM. However, additional reactive processes are not affected by any requirements, enabling the consideration of reaction networks of arbitrary complexity using PFLOTRAN's flexible input file. For example, additional elemental cycles such as Mn redox processes or inorganic C interactions with soil minerals could be added to ELM simulations with minimal changes to ELM code. From a software design perspective, ELM stores all state variables of the geochemical problem (e.g., C and N concentrations as well as aqueous concentrations of $H^+$, $SO_4^{2-}$, $HS^-$, etc. and soil minerals such as iron oxides and iron sulfides if defined in the reaction network) but only those that are directly




relevant to ELM state are visible to other model components (primarily organic matter and nutrient pools) while others are only handled by the interface. This allows for minimizing changes in ELM. We can, however, envision that certain process models in ELM could be improved on or refined with the addition of variables from the geochemical model currently not considered. For example, vegetation responses to phytotoxic sulfide concentrations or soil oxygen concentration could be added to improve representation of wetland vegetation (LaFond-Hudson and Sulman, 2023).

## 5.3 Reactive transport for integrated surface-subsurface hydrology

There is an increasing interest in using integrated hydrology models to quantify not only the water exports but also the solute exports from watersheds that impact human activities downstream (Bao et al., 2017). While integrated hydrology models have been used extensively to capture the feedback between flow in the surface and subsurface, solute transport and reactive processes are not represented in most models.

Molins et al. (2022) developed an approach to simulate reactive transport processes in integrated surface-subsurface hydrology problems. This approach was implemented in the open-source Advanced Terrestrial Simulator (ATS) (Coon et al., 2019), an integrated hydrology code built upon Amanzi solvers for subsurface flow and transport. The approach included geochemical processes both in the surface and subsurface compartments. Alquimia facilitated the implementation because ATS is built upon Amanzi, which already implemented Alquimia. In turn, the flexible multiphysics framework in ATS, which specifies interfaces for coupled processes (process kernels) and automates coupling strategies, allowed for defining separate process kernels for geochemistry in the surface and the subsurface (Fig. 5).

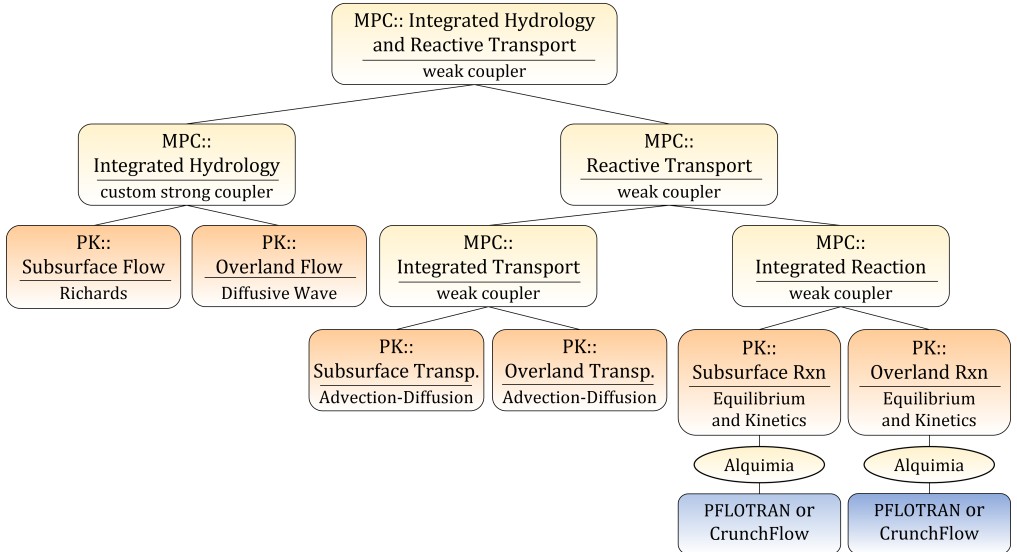

**Figure 5.** Process tree for a model of integrated hydrology and integrated reactive transport implemented in ATS (Coon et al., 2019; Molins et al., 2022) with calls to Alquimia, including both CrunchFlow and PFLOTRAN. In the example presented here, CrunchFlow is used in the Subsurface Reaction PK (left), and PFLOTRAN in the Overland Reaction PK (right). Reprinted from Molins et al. (2022).





The separation of multiphysics process kernels in ATS offers an excellent opportunity to showcase the flexibility that Alquimia provides in order to develop increasingly complex conceptual models and facilitate its implementation in software. This is exemplified here with simple simulations of reactive transport in a vertical column. To do so, we consider distinct geochemical models in the surface and subsurface, and then use both geochemical engines, CrunchFlow and PFLOTRAN, in the same simulation.

In the column, the subsurface soil is initially partially saturated and the surface is dry. Over the initial 105 days, a constant precipitation rate is applied to the surface domain, which exceeds the rate of drainage at the bottom of the column. As a result, water infiltrates into the subsurface, which becomes progressively saturated. The surface domain remains dry initially, but eventually the subsurface becomes fully saturated and water accumulates in the surface as well. The saturated hydraulic conductivity of the soil is larger than any of the prescribed water fluxes, precipitation or drainage. At 105 days, precipitation ceases and drainage increases. As a result, the ponded depth of water accumulated on the surface decreases. For the geochemical problem, we consider the presence of calcite and a set of dissolved species that describe the calcium carbonate system, including $Ca^{2+}$, $HCO_3^-$, and $H^+$ as primaries. The water in the column is initially in equilibrium with calcite. Rainwater is equilibrated with atmospheric $CO_2$, resulting in a relatively low pH, which drives calcite dissolution.

We consider 2 separate scenarios that represent two end-member conceptual models. Because in ATS surface water is treated with the shallow water approximation, water on the surface is well mixed. If heterogeneous reactions such as calcite dissolution are considered, this implies that the entire water column volume is in contact with mineral surfaces. This may overestimate the actual rate of dissolution if well mixed conditions are not achieved (e.g. gradients in concentrations exist along the water column). In scenario 1, surface water is assumed not to be in contact with the mineral at all and calcite dissolution is not included. In scenario 2, calcite is also present in the surface for dissolution.

In these simulations, we further demonstrate Alquimia's flexibility by using it to couple ATS with PFLOTRAN in the surface and CrunchFlow in the subsurface. These simultaneous couplings are not strictly necessary here because the capabilities to describe mineral dissolution and aqueous complexation in PFLOTRAN and CrunchFlow are very similar (yielding to the results shown in Section 4.3). However, one can envision situations where different engines have different capabilities, or that different process representations could be used in different parts of the domain with different codes. For example, a land surface model could be used to describe geochemical processes in the shallow subsurface in connection with SOM, vegetation and microbial dynamics, and a specialized geochemical model could be used to describe deeper subsurface processes, including mineral weathering.



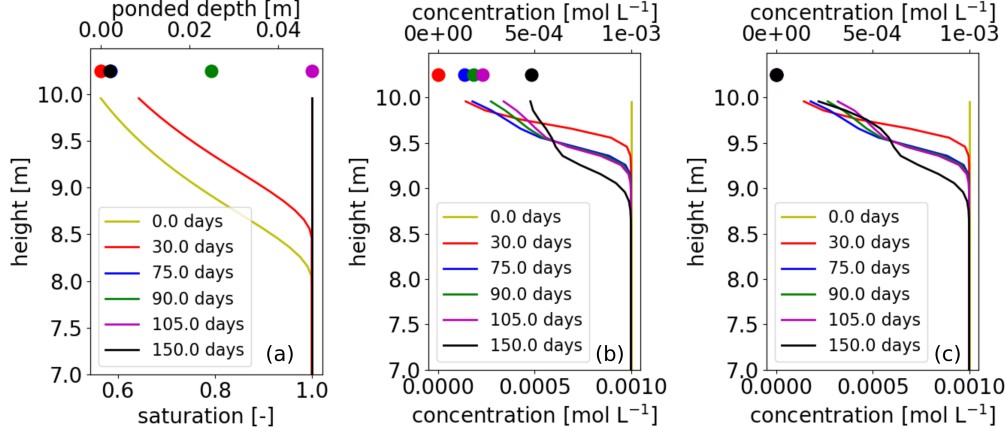

**Figure 6.** Results from surface-subsurface reactive transport simulation by ATS, with PFLOTRAN solving the geochemical problem in the surface and CrunchFlow in the subsurface, showing (a) ponded depth and water saturation, (b) $Ca^{2+}$ concentrations in scenario 1 with dissolution in the surface, and (c) $Ca^{2+}$ concentrations in scenario 2. Surface variables are shown as symbols at an arbitrary height and subsurface variables as solid lines as a function of height.

Results show differences in $Ca^{2+}$ concentrations with depth in the two scenarios (Fig. 6). Before day 75, the surface is dry and there are no solutes in the surface. Rainwater infiltrates directly into the subsurface and in both scenarios results are identical for concentrations, with concentrations increasing with depth as calcite dissolves. From day 75, concentrations appear in the surface. In scenario 1, dissolution in the surface results in an increase of concentrations there with respect to rainwater values, while in scenario 2, the values reflect those in rainwater. As surface water infiltrates, it continues to drive calcite dissolution in both scenarios. Concentration profiles until day 105 are visually very similar in both scenarios, although numerical values differ by as much as 6%. This is due to how ATS calculates mass fluxes at the surface-subsurface interface (Molins et al., 2022). As indicated in the PK tree in Fig. 5, solute fluxes are obtained first for integrated transport, then the resulting concentrations from these fluxes are used to solve the geochemical problem. In this case, this means that fluxes into the subsurface not only reflect concentrations in the surface but also concentrations in the rainwater. Because concentrations in the rainwater are essentially zero, the solution infiltrating in the subsurface until day 105 is much more diluted than that in the surface. When precipitation ceases, the infiltrating concentrations reflect solely those in the ponded water and thus start to differ more clearly between scenario 1 and 2 (Fig. 6). This type of conceptual choices are made in the development of the driver code (ATS in this case) and enabled by the flexibility of the software interface.

## 6 Discussion and conclusions

### 6.1 Flexibility

The examples presented above highlight the flexibility of the generic interface Alquimia to expand the capabilities of multi-physics simulators to include multicomponent geochemical processes. Some of these applications go beyond the use initially





envisioned for the interface, i.e. flow and reactive transport in subsurface porous media (Moulton et al., 2011). From this point of view, the operator $L$ in Eq. 2 can be seen as a generic process operator that applies to concentration and affects the mass balance of component $i$.

For example, Alquimia can be used even when there is no transport to explore the in situ dynamics of land surface processes or compare with batch-scale laboratory experiments while using driver code to expand on the native capabilities of reaction engines (Sulman et al., 2022, 2020). Further, when column-based land surface models such as ELM are used (Sulman et al., 2024), in addition to solute transport, the $L$ operator can also include gas transport and lateral fluxes associated with tidal fluctuations along the depth of column with consideration of the corresponding salinity gradients.

Generally, however, the operator $L$ includes transport processes. In the pore-scale application, transport was considered in the aqueous phase only (Sec. 5.1). In the integrated hydrology application (Sec 5.3), the subsurface was simulated as a porous medium like in Amanzi or Parflow (Sec. 4.2) but the surface was represented as a 2D domain using the shallow water approximation. There, the solute mass balance is solved for the ponded depth of water. Because the surface may be wet or dry as a result of the dynamic conditions driven by rain events, the reactive transport processes are solved only for the wet portion of the surface.

The flexibility is also demonstrated by its application to different meshing and discretization schemes. The single-cell approach enabled the use of structured and unstructured meshes in Amanzi (Moulton et al., 2011). The unstructured capabilities in Amanzi allowed for traditional finite volume schemes, mimetic finite differences, and nonlinear finite volumes (Moulton et al., 2011), while the structured capabilities allowed for adaptive mesh refinement (AMR) Dubey et al. (2014).

### 6.2 Prototyping and benchmarking

Alquimia allows for rapid prototyping new capabilities and approaches. Examples include time stepping schemes or implementations of processes that are not readily available in engine codes. In the example of Sulman et al. (2022), this enabled prescribing the oxic/anoxic fluctuations of the system from the Python driver code by exchanging oxygen via a time-varying external boundary condition. Similarly, rate constants were updated by the driver at specified time points using the fine-grained access to the reaction parameters provided by Alquimia's hands-on mode.

As a generic interface, once Alquimia is implemented in a driver, one can swap engines for the same problem as long as these engines provide the same capabilities. This is of particular interest in benchmarking of reactive transport models. Because of the coupled nature of RTM, it is important to narrow down the source of discrepancies between codes. By sharing the same engine or by sharing the same driver, the use of Alquimia allows to isolate the potential sources of discrepancies. The example presented in Sec. 4.3 demonstrated how this approach can be used to investigate the impact of discretization schemes but this could be extended to other aspects.

### 6.3 Limitations and Future work

As a generic interface, Alquimia is designed to allow for the implementation of different engines. However, the current number of implemented engines is still limited. As a result, only a small portion of all potential options have been demonstrated here.





Alquimia allows for the canonical approach (Eqs. 2-3), commonly used in reactive transport models such as PFLOTRAN
and CrunchFlow (Lichtner, 1991), but this is not required. If this approach is not available in a given engine or equilibrium
may not be assumed, there are no restrictions in setting the number of aqueous complexes ($N_x$) to zero and using the to-
tal mobile concentrations to hold all the species concentrations. Heterogeneous reactions present a similar case. For mineral
dissolution-precipitation, mineral volume fractions and surface areas are passed between driver and engine but there is no
implied assumption whether they are implemented as equilibrium or kinetic reactions in the engine. Likewise, only sorbed
concentrations are required for surface complexation reactions, which have been modeled both as equilibrium (Steefel et al.,
2015) and as kinetic processes (Greskowiak et al., 2015). The assumption on the driver side is only that total mobile concen-
trations are to be transported with the aqueous phase, while total immobile concentrations are not.

Alquimia currently only permits operator splitting coupling between engines and drivers. This poses a constraint on the time
step size and thus hampers the solution of certain problems, e.g. rapid transport. However, the interface can be expanded to
820 allow for the global implicit approach. The same single-cell model used for operator splitting would be applicable, but the
Alquimia state variables would have to include the Jacobian matrix of the geochemical problem in each grid cell. The driver
code would set up a solve the nonlinear problem resulting from coupling solute transport and reactions over the spatial domain.
For the operator splitting coupling, there are no restrictions on how it is implemented and other schemes could be considered
such as Strang symmetrization.

In the applications presented here, the choice of engine was often a function of the familiarity of the users with one of the
engines available but also with the availability of specific capabilities from one engine. While most codes with geochemical
capabilities share a set of basic capabilities (Steefel et al., 2015), one can anticipate that, as the number of engines connected to
Alquimia grows, specialized capabilities will vary more widely between engines and this will open up the range of applications
Alquimia can be used for. The example in Sec. 5.3 can serve an example of this, where different geochemical engines can be
used in different parts of the domain as appropriate. Further, increasingly geochemical models make use of machine learning
tools to accelerate calculations (Leal et al., 2020) or replace process-based calculations (Chang et al., 2023). These machine
learning models could be seamlessly integrated in the interface as additional engines, which may be used in isolation or in
combination with process-based engines as needed. While the two engines currently available are Fortran codes, the interface
is prepared to couple with engines in other languages, e.g. PHREEQC, written in C++.

Against the backdrop of continuous evolution and development of multiphysics simulators driven and enabled by new
approaches and capabilities, tools like Alquimia that simplify coupling by codifying a clear and flexible interface between
codes and processes are increasingly valuable. Specifically, Alquimia enables access to existing true-and-tried geochemical
models but also it facilitates future development and implementation of new models. Code interoperability gives access to a
range of capabilities with simple, easily maintainable code bases that also facilitate prototyping and validation of new software.
Ultimately, improved software productivity and sustainability have the potential to increase the pace of scientific discovery and
promote more efficient and effective use of research resources.



*Code availability.* Alquimia (Andre et al., 2013) is developed and maintained using the repository at github.com/LBL-EESA/alquimia-dev, with releases generally timed with xSDK releases (Bartlett et al., 2017). Different versions of the code were used for results presented here. Snapshots of Alquimia, PFLOTRAN, CrunchFlow, Amanzi, Parflow and ATS for the versions involved in this work (as noted below) as

well as the input files for the included simulations are available at dx.doi.org/10.5281/zenodo.11414442 (Molins et al., 2024a). The Amanzi source code used in this work is the version with hash 4867af7, which includes input files for Section 4.3 and reference solutions for CrunchFlow and PFLOTRAN. The Parflow source code used in this work is the version with hash d4b20b9. The versions of Alquimia, PFLOTRAN, CrunchFlow and PETSC used with these versions of Amanzi and Parflow are 1.0.9, 3.0.2., 906e164, and 3.16.0, respectively. The crunchFOAM application is described in detail by Li et al. (2022) and is based on version 1.0.6 of Alquimia. The land surface model

applications are described in detail by (Sulman et al., 2022; Wang et al., 2024; LaFond-Hudson and Sulman, 2023; Sulman et al., 2024), and are based on version 1.0.8 of Alquimia, which can be found in the model-data archives associated with these publications Sulman et al. (2020, 2023). The integrated hydrology application with ATS is described in detail by Molins et al. (2022), with results presented here obtained with the version with hash 37a7b6e of ATS, which include the input files of the simulations as part of the regression tests. The versions of Alquimia, PFLOTRAN, CrunchFlow and PETSC used in this version of ATS are 1.0.9, 5.0.0, cf938c8, and 3.20.0, respectively.

*Author contributions.* S.M. wrote the manuscript, developed Alquimia, Amanzi-ATS, implemented CrunchFlow in Alquimia, and performed the simulations in Sec. 4.3, 5.3. B.J.A. had the original idea for, designed and developed Alquimia, implemented PFLOTRAN in Alquimia, and developed PFLOTRAN. J.N.J. developed Alquimia, including compliance to xSDK policies, and implemented Alquimia in Amanzi. G.E.H. developed Alquimia, PFLOTRAN, and implemented PFLOTRAN in Alquimia. B.N.S. developed ELM-PFLOTRAN and Alquimia, including its Fortran and CFFI interfaces. K.L. developed Amanzi-U and Alquimia in Amanzi. M.S.D. developed Amanzi-S and Alquimia

in Amanzi. J.J.B. implemented Alquimia in PARFLOW. D.S. developed Amanzi-ATS and Alquimia in ATS. H.D. led the development of CrunchFOAM. P.C.L. developed PFLOTRAN, and participated in the initial Alquimia design. C.I.S. developed CrunchFlow, and participated in the initial Alquimia design. J.D.M. developed Amanzi-ATS, and led the projects that funded this work.

*Competing interests.* The authors declare no competing interests.

*Disclaimer.* Any subjective views or opinions that might be expressed in the paper do not necessarily represent the views of the U.S.

Department of Energy.

*Acknowledgements.* This material is based upon work supported as part of the IDEAS-Watersheds project funded by the U.S. Department of Energy, Office of Science, Office of Biological and Environmental Research (Contract No DE-AC02-05CH11231). Initial development of Alquimia was funded by the U.S. Department of Energy, Office of Environmental Management (Contract No DE-AC02-05CH11231). B.N.S. is supported by the U.S. Department of Energy Office of Science Early Career Research program as part of research in Earth System

Model Development within the Earth and Environmental Systems Modeling Program, and by the Next Generation Ecosystem Experiments



(NGEE) Arctic project through the Office of Biological and Environmental Research in the U.S. Department of Energy Office of Science. Oak Ridge National Laboratory is managed by UT-Battelle, LLC, for the U.S. Department of Energy under contract DE-AC05-00OR22725. PNNL is operated for the DOE by Battelle Memorial Institute under contract DE-AC05-76RL01830.





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
