# Peer review of "Alquimia v1.0: A generic interface to biogeochemical codes– A tool for interoperable development, prototyping and benchmarking for multiphysics simulators"

_Geoscientific Model Development, 2024_

## Referee Comment (RC1)

Revies comments to manuscript gmd-2024-108 by S. Molins et al., entitled "Alquimia v1.0: A generic interface to biogeochemical codes– A tool for interoperable development, prototyping and benchmarking for multiphysics simulators"

Abstract:  not clear what is a "call signature"

Introduction: "We present..." is not a good style for a scientific publication, rather "it is presented" should be used

L 65:  $c_i$ is the concentration of species $i$ per unit volume of water?

L 75: there is no $N_c$ in Eq (2). Is $N_c$ the number of primary species implicitly?

L 90: ... does not stipulate  any specific mathematical form that is used. ...

L 114: Eqs (7)

L 117: Eqs (-7)

L 133-135: What does it mean "enforcing a signature for geochemical subroutines using a single-cell model"? Why "signature" in this context? Is this a metadata structure? This needs more explanation...

L 183: Would it be nice to provide a link to GitHub repository (https://github.com/LBL-EESA/alquimia-dev) here?

L 215 on: Would it be nice to point at "call signature" or "signature for geochemical subroutines" in code block 1 or 2, to give the reader some idea of what it really is?

P 19, Figure 2 caption: in the last 3 lines, the text regarding Langmuir and Freundlich isotherms seems to miss something, e.g. "The Langmuir and Freundlich sorption isotherms presented for CrunchFlow as they are not directly available in CrunchFlow, ..."

L 653: what can be non-zero? (R_i_n from eq 12?)

L 779: The statement "Alquimia can be used even when there is no transport" sounds like a paradox in the context of Multiphysics and needs more explanation.

---

## Author Response (AR1)

**Reviewer #1**

This is a long-awaited publication - the Alquimia was released on github in 2013 and since then used in several DOE and other funded projects. The main purpose of Alquimia interface is to provide a seamless coupling between different multiphysics simulators and different chemical speciation solvers. For this, Alquimia offers a number of C data structures and API calls, sufficient for combining at least the codes written in C, C++ and Fortran. A partial Python API is also available. In principle, an Alquimia interface module needs to be created for each multiphysics code and each chemical speciation code. Currently, only CrunchFlow and PFLOTRAN chemical speciation solvers can be connected.  A particular strength of this paper is that the concepts of Alquimia are explained in several examples, from typical reactive transport up to land surface hydro-biogeochemical models. The latter couplings actually broaden the context of coupled simulations and corroborate the innovativeness of the concept and approach behind Alquimia. Overall, Alquimia - the generic interface for coupling multiphysics with geochemical codes - marks a milestone and outlines state of the art, while showing a lot of potential for future extensions e.g. with more chemical speciation solvers. This is why the work presented by S. Molins and colleagues deserves the top mark in any sense. The attached pdf contains technical comments that, if the authors choose to implement, may improve the overall quality and impact of the paper.

We thank the reviewer for the kind comments and would like to take the opportunity to provide a short development history of Alquimia that both justifies the timing of this submission as well as the role of the perspective provided in the submission. In a sense, this contribution is a product of this long process.

It was initially conceived and developed as a solution to provide proven geochemical capabilities to the code Amanzi during its development phase, specifically using PFLOTRAN geochemical subroutine routines. The advantages of the generic stand-alone design, rather than being tied exclusively to PFLOTRAN, became apparent when CrunchFlow was added to the interface. As such, it became a representative example of code interoperability, at a time where lack of interoperability had been identified as an obstacle for code development in the Department of Energy (DOE) software ecosystem (from linear solvers to application codes). In developing Amanzi's test suite, it also became apparent the possibilities that swapping drivers and engines opened up for code intercomparison efforts under way (Steefel et al., 2015) and references therein). Over the last few years, in addition to maintaining and refining aspects of the interface, most work has gone into implementing it into additional codes. In one case, it was done to replace a previous coupling to CrunchFlow; in another case, the availability of a well-defined and documented interface facilitated implementation into land models; in a third, the availability of a C-interface and implementation examples in the test drivers facilitated coupling to openFOAM, and lastly, ATS relied on the earlier Amanzi implementation due to their close relationship. While it was indeed a long-time coming, it would have not been possible to envision and present all these examples and applications back in 2013.

The technical comments provided in the pdf helped us improve the manuscript by clarifying some terms that are not clear in the draft, such as call signature, and correct a number of errors. They are detailed in what follows:

Abstract: not clear what is a "call signature"

Call signature refers to the function arguments and their types, whether input or output. We have replaced it by a more explicit expression, avoiding the use of the word signature:

"The interface enforces the function arguments and their types for setting up,..."

Introduction: "We present…" is not a good style for a scientific publication, rather "it is presented" should be used

Yes, the sentence is unnecessarily long. We simplify it to

"Alquimia v1.0 is a generic interface to geochemical solvers that facilitates development of multiphysics simulators by enabling code coupling, prototyping and benchmarking."

L 65: ci is the concentration of species i per unit volume of water?

Yes:

"ci is the concentration of species i (mass per unit water)"

L 75: there is no Nc in Eq (2). Is Nc the number of primary species implicitly?

Yes, we specific the number of equations in each case, Eq. (1)-(6).

L 90: … does not stipulate  any specific mathematical form that is used. …

We rephrased it

"does not stipulate any specific mathematical form."

L 114: Eqs (7)

"Eq(7)"

L 117: Eqs (7)

"Eq(7)"

L 133-135: What does it mean "enforcing a signature for geochemical subroutines using a single-cell model"? Why "signature" in this context? Is this a metadata structure? This needs more explanation…

We replace the term and provide an explanation that clarifies what we mean:

"Alquimia has two parts: (1) an engine-independent application programming interface (API) consisting of all relevant functions, data structures, constants, and their respective types (Table 1) and (2) an optional utility library."

L 183: Would it be nice to provide a link to GitHub repository (https://github.com/LBLEESA/alquimia-dev) here?

" has an accessible repository (https://github.com/LBL-EESA/alquimia-dev, 2024)."

L 215 on: Would it be nice to point at "call signature" or "signature for geochemical subroutines" in code block 1 or 2, to give the reader some idea of what it really is?

As noted earlier, we are using different wording. When referring to these code blocks at line 277, we write:

"While Alquimia's approach to engine data sometimes introduces more detail in the code, its flexibility allows Alquimia to accommodate the needs of very different engines. Ultimately, this makes it possible for example for the subroutine ReactionStepOperatorSplit to share the same arguments for different engines (compare lines 1 and 2, code blocks 1 and 2)."

P 19, Figure 2 caption: in the last 3 lines, the text regarding Langmuir and Freundlich isotherms seems to miss something, e.g. "The Langmuir and Freundlich sorption isotherms presented for CrunchFlow as they are not directly available in CrunchFlow, …"

We correct it to:

"The Langmuir and Freundlich sorption isotherms are not presented for CrunchFlow. While they can be simulated via a surface complexation model and a single sorbing species, no specific keyword in the input deck is available and this was not pursued further here."

L 653: what can be non-zero? ($R\_i\_n$ from eq 12?)

Yes:

", $R^n_{i}$ can be non-zero (i.e. dissolution or precipitation may take place)."

L 779: The statement "Alquimia can be used even when there is no transport" sounds like a paradox in the context of Multiphysics and needs more explanation.

The only thing that this means is that there is no requirement when using Alquimia to include any and all processes in a multiphysics simulation or be solving a spatially

distributed problem. In the same way, that e.g. one can use PHREEQC, PFLOTRAN or CrunchFlow as batch simulators. It was stated in the wrong way. This is rephrased to make it clearer

"For example, Alquimia can be used to perform simple batch chemistry calculations of the kind routinely carried out by geochemical models such as PHREEQc, without consideration of other processes that involve fluxes over a spatially discretized domain. This may be necessary for batch-scale laboratory experiments or, as in the land surface model example, to expand the range of reactive processes considered (Sulman et al., 2022, 2020)."

**Reviewer #2**

This excellent paper introduces Alquimia, a software serving as a generic coupling interface to facilitate the coupling of geochemical simulators codes into multi-physics frameworks.

The authors start by describing the software landscape of reactive transport models (RTM), including a formulation of the governing equations for geochemistry (in terms of primary species, with secondary species linked to them by means of mass action laws). The role of the alquimia library is to sit between the "driver" and the "engine". The driver is the transport code which sets up the spatial discretization and defines the processes that are simulated; the engine is the goechemical solver. The user needs to implement both the alquimia-interface for the driver and the engine interface; the latter is conveniently driver-agnostic, meaning that once an alquimia-interface for the geochemical engine is available, it can be reused by any driver.

Alquimia itself is developed in C for maximum portability and its API can be easily called by C/C++ and fortran, which are the most common languages used to develop multiphysics applications and RTM in general.

Section 4 continues with implementation examples from already developed interfaces in case of engines PFLOTRAN and CrunchFlow, which are inherently different in terms of expected data structures and auxiliary variables, highlighting the flexibility and added value in terms of maintainability and extendability provided by the alquimia approach. Notably, examples of coupling code dealing with adaptive grid refinement and with different modi for the same driver (amanzi, structured grids or unstructured grids) are provided and discussed to explain important design choices for alquimia.

Finally, section 4.3 presents results of 8 different combinations of drivers and engines for the same problems, highlighting the fact that once the alquimia interfaces are implemented, the user can transparently test different codes and gauge the differences in results due to the different numerical schemes for transport and the actual engines.

Section 5 presents further high-level use case for alquimia, pertaining the coupling of geochemical engines to unconventional reactive transport models, showcasing the ease of prototyping such a coupling using the alquimia framework, notably using two different engines in different domain regions in the ATS simulation.

Finally the paper closes discussing current limitations and future work. The paper is excellent and the presented software innovative and I have just minor comments for the authors:

We thank the reviewer for the kind comments and we are glad to see that the messages that we wanted to convey came through in his/her reading as well. We address the comments below to improve the article.

- It is correctly stated that chemistry is embarassingly parallel and that alquimia is designed with a "single cell" approach which makes it possible to flexibly deal with hardware resources, however no further comment nor code example is given in the paper about it. I would like the authors to elaborate on this aspect since it is in my opinion crucial in the scope of RTM. Is the parallelization strategy completely determined by the driver (e.g., reusing domain partitions as the AMR example of code listing n. 4 seems to implicitly suggest) or does alquimia explicitly allow to introduce ad-hoc parallelization schemes for the geochemical engine in the coupling interface? As an exemplary use case, the driver (transport code) would use N domain partitions on N CPUs to solve flow and transport, but chemistry could be solved on M > N CPUs since linear scaling is expected for geochemistry. Could for example a round robin parallelization independent on the one used in the transport code be implemented at the alquimia level, as e.g. implemented by https://doi.org/10.5194/gmd-14-7391-2021 ?

Yes, we have added a discussion on Parallelization in the discussion. Specifically, the new Section 6.3, which provides a response to this comment

"6.3 Parallelization

Alquimia is a single-cell or 0-dimensional model that has no notion of the spatial problem. As discussed in Sec. 5.1 and Sec.5.3, this assumes well-mixed geochemical conditions in this cell, which can be viewed as a batch reactor. The geochemical problem can be obtained independently from all other cells that make up the discretized spatial domain in a multiphysics problem. Hence, the driver determines the parallelization strategy for the solution of the spatially distributed, multiphysics problem. Because the geochemical equations are uncoupled across the domain, there is considerable freedom in staging their integration and load balancing.

All code examples presented here, except the python-based prototype in Sec. 5.2, are multithreaded CPU-based implementations and involve Message Passing Interface (MPI) parallelization by the driver. The work to integrate all the cells in the domain can be distributed across the available threads with no race or synchronization concerns. Load balance can be achieved by evenly distributing the work across processors, and within each processor across available threads. Code block 4 shows how Amanzi-S is explicitly exploiting multi-threading within a Fortran Array Box. However, the load balancing of chemical calculations may be at odds with load balancing strategies that incorporate stencil operations (such as advection or diffusion).

While none of the examples presented tested this, it is possible to use different load balancing strategies for transport and chemistry when using Alquimia. The driver could choose to redistribute the state data between each stage of the time-split integration differently, including one that seeks better load balancing when sharp geochemical fronts are present in areas of the domain (and thus computations are more expensive) such as round-robin approaches, e.g. (De Lucia et al., 2021). The situation is more complex in GPU implementations (Balos et al., 2025). This contribution could not address the GPU use case with the capabilities available in the engines and drivers using Alquimia.

Geochemical engines could also implement a form of task parallelism for the geochemical solution within each cell. Although it is not the case for PFLOTRAN or CrunchFlow, it has been suggested that such a form of parallelism could speed up geochemical calculations in systems with a very large number of species and reactions. Anecdotal reports indicate that this has been effective for the codes Geochemist's Workbench (GWB) ChemPlugin (Bethke, 2024) and TOUGHREACT (Sonnenthal et al., 2021).

- Formal references to eqs solved by Alquimia and geochemical engine respectively are to be checked: sometimes (5-7), e.g., lines 114, 117, but often (8-5), e.g. lines 140, 142. Please carefully review these references to avoid confusion.

Agreed. We have fixed the reference to these equations so that it is consistent, and simpler.

Line 115

"The Alquimia interface is designed to act as a generic, intermediary layer between a code that solves Eq. (7) and a code that solves Eq. (8). We will refer to the former code as the driver and the latter as the engine (Fig. 1).

The driver is the code that drives the simulation, handles the spatial description of the problem, including the meshing and spatial discretization, and solves Eq. (7)."

Line 140

"to obtain initial and boundary concentrations as needed by the engine, solving a steady-state form of Eq. (8), with additional constraints such as fixed species concentration or pH values, and charge balance or mineral equilibrium. AlquimiaReactionStepOperatorSplit performs the solution of the geochemical problem (Eq. 8)."

**References**

Balos, C. J., Day, M., Esclapez, L., Felden, A. M., Gardner, D. J., Hassanaly, M., et al. (2025).

SUNDIALS time integrators for exascale applications with many independent systems of

ordinary differential equations. *The International Journal of High Performance Computing Applications*, *39*(1), 123–146. https://doi.org/10.1177/10943420241280060

Bethke, C. M. (2024). *The Geochemist's Workbench® Release 17 ChemPlugin™ User's Guide*. Champaign, Illinois: Aqueous Solutions, LLC. Retrieved from https://www.gwb.com/pdf/GWB/ChemPluginUsersGuide.pdf

De Lucia, M., Kühn, M., Lindemann, A., Lübke, M., & Schnor, B. (2021). POET (v0.1): speedup of many-core parallel reactive transport simulations with fast DHT lookups. *Geoscientific Model Development*, *14*(12), 7391–7409. https://doi.org/10.5194/gmd-14-7391-2021

Sonnenthal, E., Spycher, N., Xu, T., & Zheng, L. (2021). TOUGHREACT V4.12-OMP and TReactMech V1.0 Geochemical and Reactive-Transport User Guide. Retrieved from https://escholarship.org/uc/item/8945d2c1

Steefel, C. I., Appelo, C. A. J., Arora, B., Jacques, D., Kalbacher, T., Kolditz, O., et al. (2015). Reactive transport codes for subsurface environmental simulation. *Computational Geosciences*, *19*(3), 445–478. https://doi.org/10.1007/s10596-014-9443-x